# RNA demethylase *FTO* participates in malignant progression of gastric cancer by regulating SP1-AURKB-ATM pathway
**Xueliang Zeng**[1,2,4], **Yao Lu**[1,3,4], **Taohui Zeng**[2], **Wenyu Liu**[2], **Weicai Huang**[2], **Tingting Yu**[2], **Xuerui Tang**[2], **Panpan Huang**[2], **Bei Li** [2] ✉ & **Hulai Wei** [1] ✉

Gastric cancer (GC) is the 5[th] most prevalent cancer and the 4[th] primary cancer-associated mortality globally. As the first identified m6A demethylase for removing RNA methylation modification, fat mass and *obesity-associated protein* (*FTO*) plays instrumental roles in cancer development. Therefore, we study the biological functions and oncogenic mechanisms of *FTO* in GC tumorigenesis and progression. In our study, *FTO* expression is obviously upregulated in GC tissues and cells. The upregulation of *FTO* is associated with advanced nerve invasion, tumor size, and LNM, as well as the poor prognosis in GC patients, and promoted GC cell viability, colony formation, migration and invasion. Mechanistically, *FTO* targeted specificity protein 1 and Aurora Kinase B, resulting in the phosphorylation of ataxia telangiectasia mutated and P38 and dephosphorylation of P53. In conclusion, the m6A demethylase *FTO* promotes GC tumorigenesis and progression by regulating the SP1-AURKB-ATM pathway, which may highlight the potential of *FTO* as a diagnostic biomarker for GC patients' therapy response and prognosis.

Worldwide gastric cancer (GC) poses a critical public health challenge in that it has more than a million new cases and 769,000 deaths worldwide in 2020[1]. Despite the development in diagnosis and treatment, the outcome of advanced GC remains poor, and patients with distant metastases have a less than 5% five-year survival rate[2–4]. Therefore, identifying novel therapeutic targets and understanding the molecular mechanisms underlying GC development are critical to improving the early diagnosis of GC.

Recently, the function of RNA modifications in cancers has garnered growing research attention[5]. The most prevalent form of methylation in RNAs is N6-methyladenosine (m6A), and its dysregulation has been associated with cancer pathogenesis including GC[4,6–9]. For example, demethylase *ALKBH5* inhibits the spread of gastric cancer through moderating *PKMYT1* m6A[10]. By controlling the *miR-30c-2-3p/AKT1S1* axis, METTL14-mediated m6A alteration of *circORC5* prevents the advancement of gastric cancer[11]. Fat mass and *obesity-associated protein (FTO)* is the first identified m⁶A demethylase to specifically remove RNA m⁶A modification[12]. *FTO* is overexpressed in many cancers and has obvious impacts on the development of cancers[13,14], by regulating mRNA splicing, stability, and translation[9,15]. However, the underlying mechanisms of *FTO* in GC tumorigenesis remain a mystery.

Specificity protein 1 (SP1) is a transcription factor that enhances transcription of oncogenes during cell proliferation, differentiation, and apoptosis[16,17]. For example, *USP39* promotes the tumorigenesis of hepatocellular carcinoma by stabilizing and deubiquitinating SP1[18]. *LncRNA THAP7-AS1* promotes oncogenesis by enabling *CUL4B* entrance into the nucleus. It is transcriptionally triggered by SP1 and post-transcriptionally maintained by METTL3-mediated m6A modification[19]. The Mitotic syndicate Aurora Kinase B (AURKB) is a key regulator of mitosis and cisplatin resistance, promoting GC growth, development, and metastasis[16,17,20]. Germ-line mutations in the ataxia telangiectasia mutated (ATM) gene lead to ataxia-telangiectasia syndrome, manifested by increased susceptibility to malignancies[21,22].

The purpose of this study was to examine the function of *FTO* in GC proliferation and metastasis. We identified a novel signaling pathway involving SP1, AURKB and ATM that is necessary for GC progression, providing potential biomarkers for developing therapeutic strategies against GC.

## Results

### Upregulation of FTO is identified in GC and associated with adverse prognosis

To explore the role of *FTO* in GC development, *FTO* expression level, patient survival rates and various clinicopathological parameters were

[1]Key Laboratory of Preclinical Study for New Drugs of Gansu Province, School of Basic Medical Sciences, Lanzhou University, Lanzhou, Gansu 730000, China. [2]Department of Pharmacy, The First Affiliated Hospital of Gannan Medical University, Gannan Medical University, Ganzhou, Jiangxi 341000, China. [3]School of Basic Medicine, Gannan Medical University, Ganzhou, Jiangxi 341000, China. [4]These authors contributed equally: Xueliang Zeng, Yao Lu. ✉e-mail: Belinda0512@163.com; weihulai@lzu.edu.cn

analyzed based on the TCGA dataset and/or the local enrolled patients. *FTO* was obviously upregulated in the malignant tissues of GC patients compared to their normal tissues (Fig. 1a, Supplementary Fig. 1a). To validate this result, 20 clinical samples were collected, and *FTO* expression levels were assessed (Supplementary Table S1). Both protein and mRNA levels of *FTO* were greater in malignant tissues than in paracancer tissues as revealed by IHC (Fig. 1b), Western Blotting (Fig. 1c), and qPCR (Fig. 1d). Moreover, the survival curve showed that patients with increased *FTO* expression levels exhibited considerably worse prognoses in the first progression (FP), the overall survival (OS), and the post-progression survival (PPS) (Fig. 1e). As shown in Fig. 1f and Supplementary Table S1, the upregulation of *FTO* was markedly associated with advanced nerve invasion, tumor size and lymph node metastasis (LNM). Furthermore, the levels of *FTO*'s mRNA (Fig. 1g) and protein (Fig. 1h) expression were markedly higher in the human GC cells (HGC27, MKN-45, AGS, and SGC-7901) than in the human normal gastric mucosal epithelial cells (GES-1). Together, these data suggest that *FTO* expression is upregulated and is associated with oncogenesis and poor clinical outcome in GC patients.

### Knock-down of FTO inhibits proliferation, migration, and invasion of GC cells in vitro

To validate the function of *FTO* in GC progression, *FTO* was knocked down in AGS and SGC-7901 cells using FTO-specific shRNA. The knockdown efficiency of *FTO* was confirmed in GC cell lines by qPCR and western blotting (Fig. 2a, b). Successively, the effect of *FTO* on the proliferation, migration, and invasion of GC cells was evaluated. *FTO*-knockdown dramatically decreased the viability of GC cells, according to the CCK-8 assay (Fig. 2c). Moreover, the staining results showed that low *FTO* expression obviously reduced the colony formation of GC cell lines (Fig. 2d). Furthermore, the transwell experiment showed that lowered *FTO* expression markedly weakened the migration cells number of AGS and SGC-7901 cells (Fig. 2e). Also, the invasion capacity of the two cells was markedly inhibited by *FTO* knockdown (Fig. 2f). These evidence suggest that *FTO* can encourage the proliferating, migrating, and invading capacity of GC cells in vitro.

### SP1 is a downstream target of FTO and regulates GC progression

Having confirmed that *FTO* can facilitate the proliferating, migrating, and invading GC cells. We set out to explore the mechanisms by investigating m6A methylation using the m6A RNA methylation sequencing in AGS cells with *FTO*-knockdown and control (Fig. 3a, b). We found that the RNA methylation level of SP1 was markedly increased in the *FTO*-knockdown group. Notably, transcriptome sequencing revealed that the transcriptional level of SP1 was downregulated in the *FTO*-knockdown group. (Fig. 3c). This result indicated that the m6A demethylases *FTO* markedly suppressed methylation of SP1 mRNA and enhanced transcription of SP1. As SP1 is an important transcription factor regulating cell proliferation, differentiation, and apoptosis[16,17], we therefore evaluated the relation between the expression of *FTO* and SP1 with correlation analysis software. SP1 expression was positively associated with *FTO* expression (Fig. 3d). Furthermore, the methylated RNA Immunoprecipitation assay using m6A-specific antibody confirmed that SP1 was markedly enriched by the m6A-specific antibody in GC cell lines but the control IgG antibody did not (Fig. 3e). Moreover, *FTO*-knockdown markedly inhibited SP1 expression in both AGS and SGC-7901 cells (Fig. 3f, g). Next, we conducted the RNA pulldown-MS assay to determine the RNA methylation reader protein of SP1. We identified an obvious binding between the m⁶A reader protein, YDHTF2, and the *SP1* RNA. We conducted an RNA decay assay in GC cells treated with ActD and found that *SP1* mRNA degradation was accelerated by *FTO* silencing and stabilized after silencing YDHTF2 expression (Fig. 3h). *SP1* downregulation by *FTO* knockdown was restored by silencing YDHTF2 expression (Fig. 3i, j). Taken together, the above data suggest that FTO controls SP1 expression by removing its RNA methylation.

### SP1 is upregulated and predicts unfavorable prognostic in GC patients

To study the functionality of *SP1* in GC progression, *SP1* expression levels and patient survival rates were analyzed based on the TCGA dataset. *SP1* expression was noticeably more abundant in STAD patients' tumors than in their normal tissues (Fig. 4a, b). The survival curve showed that patients with increased *SP1* expression levels exhibited considerably worse prognoses in the OS, FP, and PPS (Fig. 4c). Furthermore, *SP1* expression levels were validated by Western blot, immunohistochemistry, and qPCR assays in the cancerous tissues of GC patients (Fig. 4d–f). Collectively, the above results indicated that SP1 upregulation is related to the tumorigenesis and poor prognosis of GC patients.

### Knockdown of SP1 inhibits the GC progression in vivo and in vitro

To better understand the functions of *SP1* in GC development, *SP1* stable knockdown cell lines were established with recombinant lentiviruses. CCK-8 assay showed that *SP1*-knockdown remarkably reduced the proliferation of AGS and SGC-7901 cells (Fig. 5a and Supplementary Fig. 1b). Meanwhile, through crystal violet staining, we found that low-expressed *SP1* obviously reduced the colony formation in GC cell lines (Fig. 5b). According to the transwell experiment, *SP1* knockdown dramatically reduced the abilities of GC cells to migrate and invade (Fig. 5c, d). Furthermore, we subcutaneously injected the AGS cells stably expressing *sh-SP1* into nude mice to construct a xenograft model of GC. Animal experiments showed that SP1-knockdown obviously inhibited tumor growth in vivo (Fig. 5e). Taken together, these results indicate that *SP1* plays a critical role in GC progression.

### SP1 promotes GC development by targeting AURKB

To deepen our understanding of *SP1*'s role in GC development, we first predicted the downstream target genes of *SP1* through the JASPAR website (http://jaspar.genereg.net/) (Fig. 6a). Then, we evaluated the candidate genes expression levels based on the TCGA dataset. *AURKB* was identified to be obviously upregulated in the tumor compared to the normal tissues of STAD patients (Fig. 6b and Supplementary Fig. 2a). The interaction between *SP1* and AURKB was then explored by determining the *AURKB* expression in the SP1-knockdown cells. We found that SP1 knockdown markedly downregulated *AURKB* expression in both AGS and SGC-7901 cells (Fig. 6c, d). Meanwhile, the high-expressed *AURKB* was confirmed in the cancerous tissues of the GC patients (Fig. 6e, f). Furthermore, the luciferase experiments showed an increased luciferase activity of *AURKB* promoter region in the AGS cells overexpressing *SP1* (Fig. 6g). In addition, FTO knockdown markedly inhibited *AURKB* expression in GC cell lines (Fig. 6h, i). The protein-to-protein interaction network (https://cn.string-db.org/, https://signor.uniroma2.it/) showed that *AURKB* could interact with ATM, P53, and P38, among others (Supplementary Fig. 2b and c). Reportedly, *AURKB* mediates ATM phosphorylation[23] and ATM plays an important role in inducing GC susceptibility[21], and the activation of P38 by ATM may be an alternative pathway to induce *p53* mutation. Therefore, we detected the phosphorylated and total proteins of ATM, P38, P53 in GC cell lines with *AURKB* or *SP1* knockdown to further explore the downstream signaling. The results showed that the *AURKB*- or *SP1*-knockdown markedly inhibited the levels of phosphorylated ATM and P38, while upregulating the level of phosphorylated P53 (Fig. 6j). Moreover, inhibition of *SP1* with its inhibitor, plicamycin, revealed similar consequences, also confirming ATM as a downstream target of *SP1* and *AURKB* (Supplementary Fig. 2d). The above results support the involvement of *SP1-AURKB-ATM* axis in regulating GC progression by targeting P38/P53.

### FTO promotes GC development by regulating the SP1-AURKB-ATM axis

To further determine whether FTO contributes to GC development by specifically regulating *SP1-AURKB-ATM* axis, we first investigated the biological functions of GC cells after *FTO*-knockdown with or without SP1/AURKB overexpression (NC, sh-FTO, sh-FTO+oe SP1, sh-FTO+oe

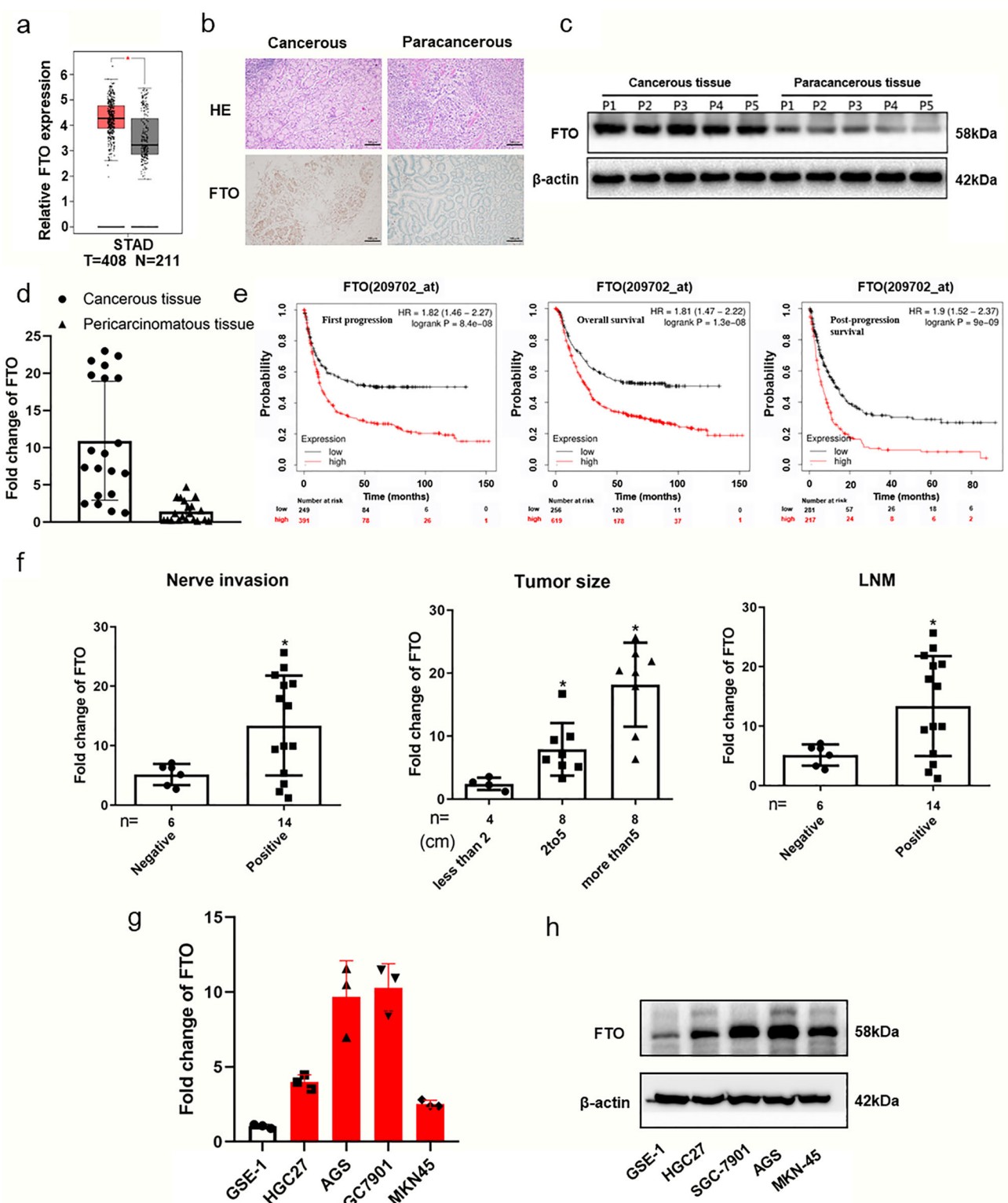

**Fig. 1 | *FTO* is upregulated and predicts poor outcomes in GC patients. a** FTO expression levels in STAD (stomach adenocarcinoma) and normal tissues from the TCGA database. **b**–**d** *FTO* expression levels in cancerous and paracancerous tissues of GC patients detected by IHC (**b**), Western blotting (**c**), and RT-qPCR (**d**) assays. **e** Kaplan–Meier curves of the first progression (FP), the overall survival (OS), and the post-progression survival (PPS) in STAD patients with high or low *FTO* expression levels. **f** Relationship between the upregulation of *FTO* in the cancerous tissues with the pathological progression of GC patients. **g**, **h** *FTO* expression levels were determined in the human GC cell lines and normal gastric mucosal epithelial cells by RT-qPCR (**g**) and Western blotting (**h**) assays. β-actin was used as an internal control for the RT-qPCR and the Western blotting assays. *$p < 0.05$, **$p < 0.01$.

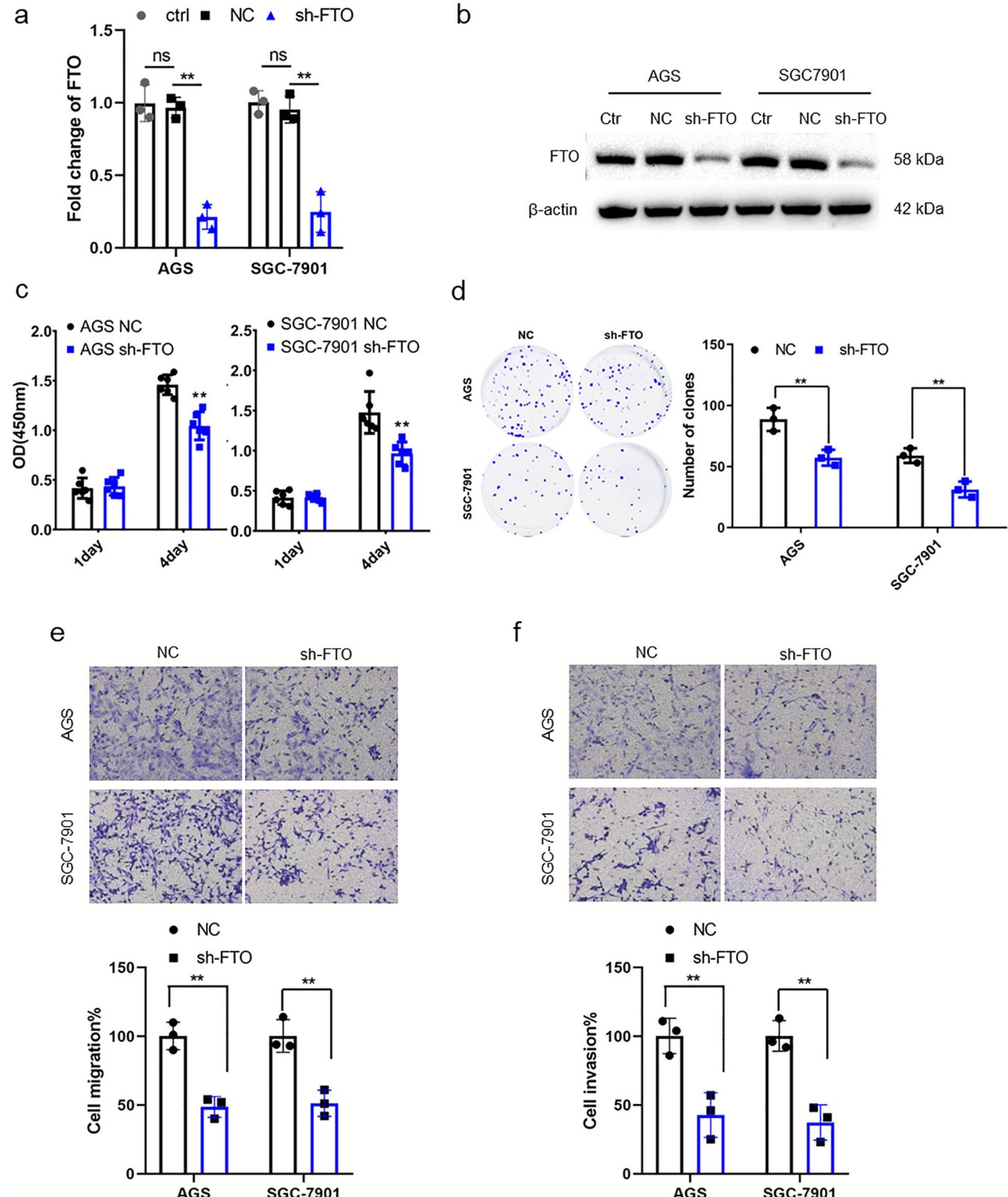

**Fig. 2 | *FTO* knockdown inhibits the proliferation, migration, and invasion of GC cells in vitro. a, b** The *FTO* expression with FTO-shRNA in GC cell lines was assessed by qPCR (**a**) and Western blotting (**b**). **c** The proliferation of GC cells tested by CCK-8 experiments. **d** *FTO* knockdown suppressed colony formation of GC cells. **e, f** Transwell assays revealed that *FTO* knockdown inhibits GC cell migration and invasion in vitro.

AURKB). Our results showed that the viability or proliferation (Fig. 7a), the colony formation (Fig. 7b), the migration (Fig. 7c) and invasion (Supplementary Fig. 2e) abilities of AGS and SGC-7901 cells were obviously inhibited by FTO-knockdown (*sh-FTO*), which could almost be fully rescued by overexpressing either SP1 (sh-FTO+oe SP1) or *AURKB* (sh-FTO +oe AURKB). Moreover, the inhibited ATM and P38 and increased P53

phosphorylation by FTO knockdown were nearly restored by SP1 or AURKB overexpression (Fig. 7d). These findings indicated that the upregulation of SP1/AURKB expression by FTO exerted a major role in driving GC development by releasing the inhibition of P53 conferred by ATM/P38.

Furthermore, the stable *AURKB*-knockdown (AURKB-KD) AGS cells were subcutaneously injected into nude mice to make a xenograft GC

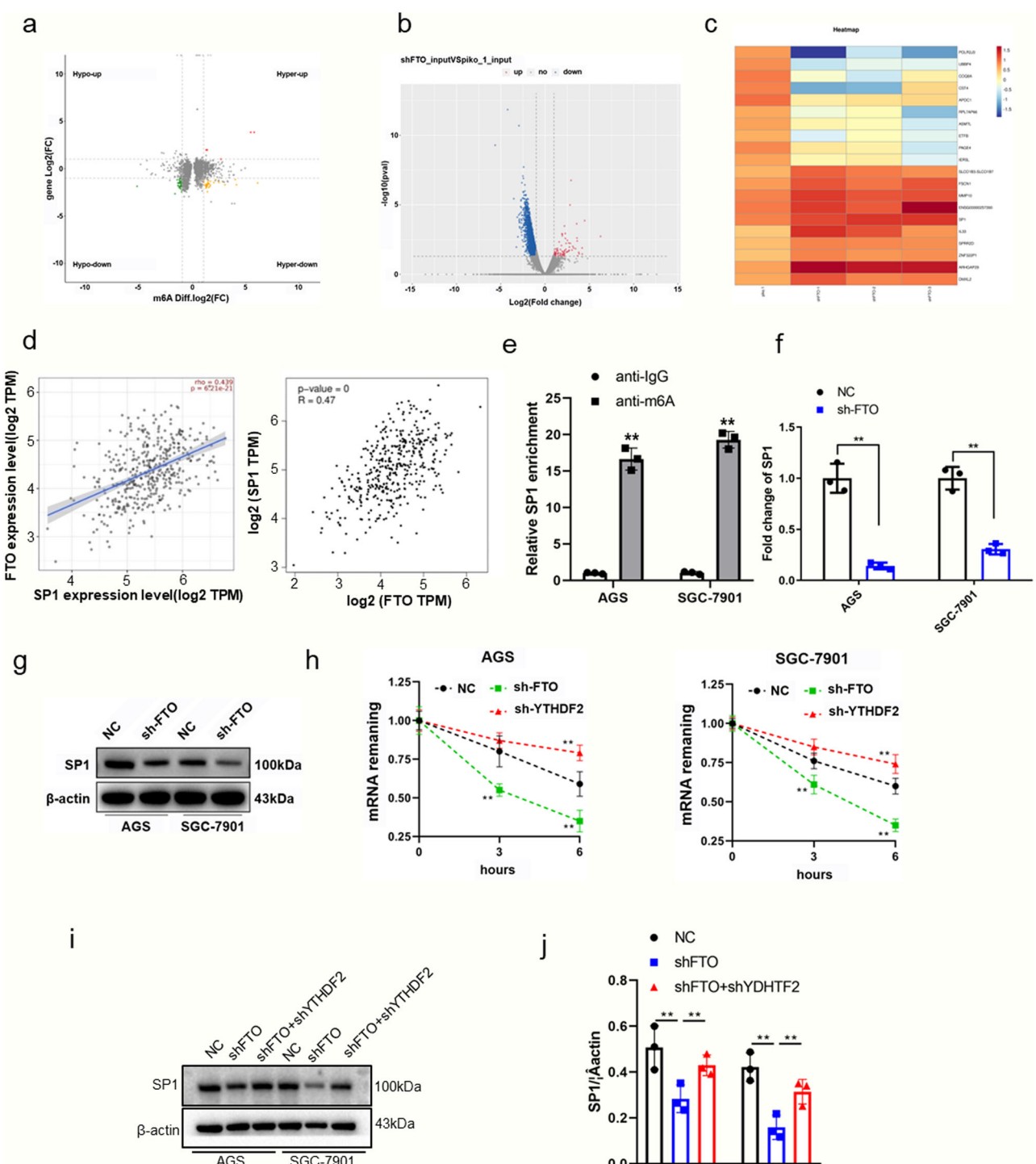

**Fig. 3 | *FTO* knockdown results in increased RNA m⁶A methylation and decreased transcription in *SP1* but is restored by silencing *YDHTF2*.**
**a** Downregulated and upregulated m⁶A peaks. **b** Volcano plots show significant differences in m⁶A peaks. **c** Heatmap of the differential genes. **d** *SP1* was recognized as the downstream target gene of *FTO* by the correlation analysis of gene expression. **e** MeRIP-qPCR assay demonstrated the *FTO* protein-SP1 RNA interaction. **f** *SP1* mRNA level was assessed by RT-qPCR. **g** The protein level of SP1 was assessed by Western blotting. **h** RNA decay assay was performed by ActD treatment in AGS cells transfected with *sh-FTO* and *sh-YDHTF2*. **i** Western blotting and **j** RT-qPCR assays were carried out to observe the changes of *SP1* in *FTO*-knocked-down GC cell lines with or without co-transfecting the *shYDHTF2*. *$p < 0.05$, **$p < 0.01$.

model. We observed that *AURKB*-KD obviously inhibited the growth of GC cells in vivo compared to the control GC-bearing mice (Fig. 7e). This result indicates that AUKRB acts as a promoting factor for GC. In an attempt to study the clinical significance of the *SP1-AURKB-ATM* axis in GC development, we utilized an SP1 inhibitor, plicamycin, to suppress SP1 expression in AGS and SGC cells and found that SP1 obviously inhibited the expression levels of *AURKB* and the levels of phosphorylated *ATM* and P38, while upregulating the level of phosphorylated P53 (Supplementary Fig. 2d). Together, these results indicate that the *SP1-AURKB-ATM* axis plays a critical role in the development of GCs (Fig. 8).

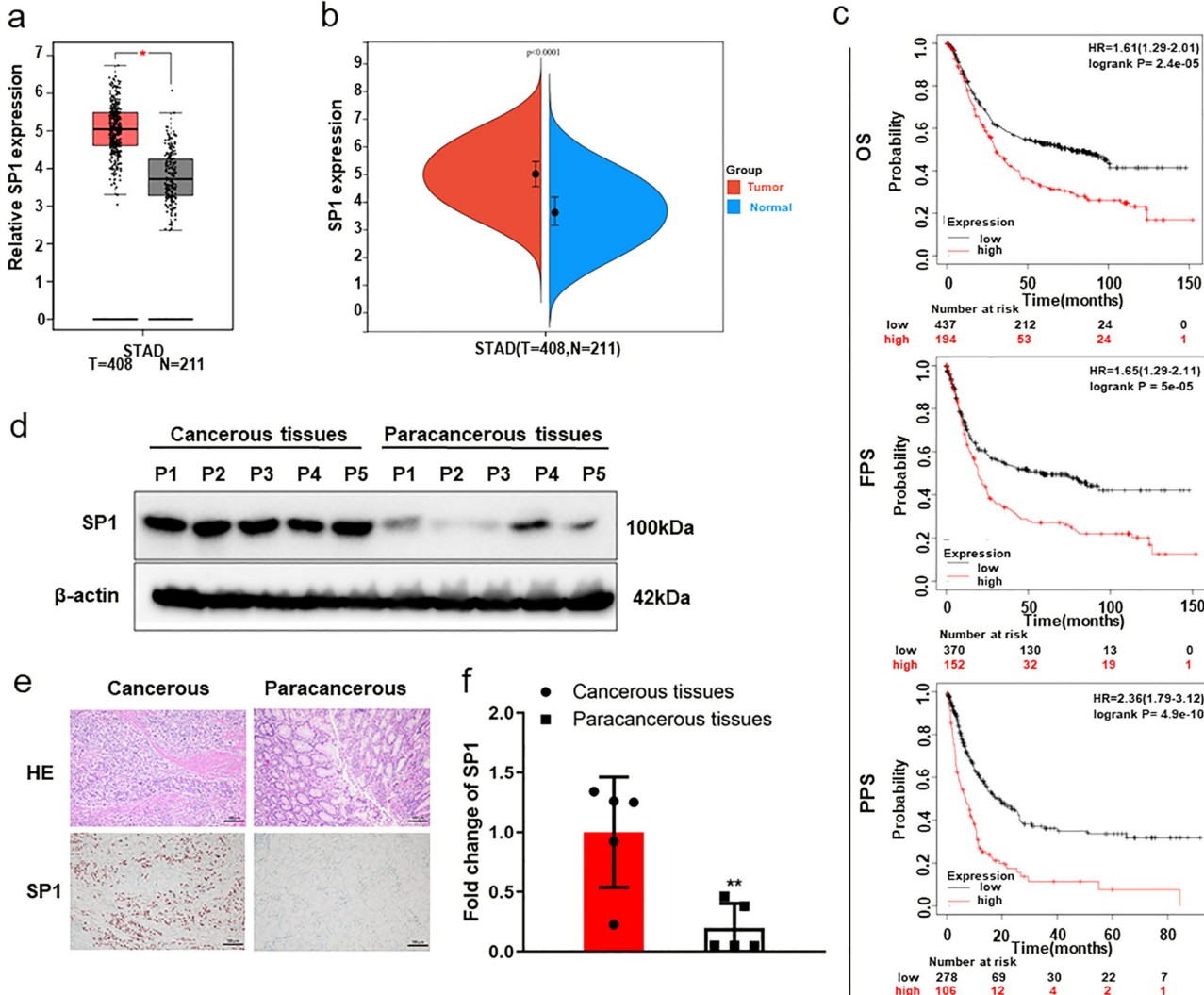

**Fig. 4 | *SP1* is upregulated and predicts poor prognosis in GC patients. a**, **b** *SP1* expression in STAD and normal tissues from the TCGA database. **c** Kaplan–Meier curves of the overall survival (OS), the first progression (FP), and the post-progression survival (PPS) of STAD patients with high or low SP1 expression. **d**–**f** *SP1* expression in the cancerous and paracancerous tissues of GC patients by Western blotting (**d**), immunohistochemical detection (**e**), and qPCR (**f**). *$p < 0.05$, **$p < 0.01$.

## Discussion

RNA epigenetics, in particular, reversible and dynamic m6A RNA methylation, has emerged as critical manner of gene expression regulation during carcinogenesis and cancer progression[24]. As the first identified m6A RNA demethylase, the significance of FTO in cancer research is growing[25]. The importance of FTO lies in its ability to regulate RNA demethylation, specifically the removal of m6A modifications[4,6,26].

Here, we focused on the role of *FTO* in the development of GC and its potential as a therapeutic target. Understanding the role of *FTO* in cancer progression is crucial for developing effective strategies to diagnose and treat GC. Hence, we performed a comprehensive analysis using a combination of bioinformatics, clinical data, molecular assays, and functional experiments. Firstly, we showed that GC tissues and cells dramatically increased *FTO*, which was associated with advanced nerve invasion, tumor size, and LNM, as well as a poor prognosis. This result is consistent with previous research that implicated *FTO* in various cancers, underscoring the clinical relevance of FTO in GC[27]. The results of RNA methylation sequencing showed that, after FTO knockdown in GC cells, the methylation state of lots of genes especially the genes related to tumor metastasis and metabolism were markedly altered, indicating that FTO may play an obvious regulatory role in the occurrence and development of GC via regulating the homeostasis of

RNA methylation and demethylation of malignant progression- or metabolism-related genes (Supplementary Fig. 3a–c). The included accurate genes or pathways in this process need further exploration. Our studies confirmed that FTO promoted GC cell viability, colony formation, migration, and invasion, providing compelling evidence for the oncogenic role of FTO in GC progression. However, according to analyzing the TCGA database, there was no significant relationship between the expression of FTO (or SP1) and the grades, stages, or subtypes of gastric cancer (Supplementary Figs. 4 and 5). Moreover, we identified SP1 and AURKB as downstream targets of FTO, unveiling a signaling axis involved in GC progression. The tumors possess a unique microenvironment to control the fate of the tumor cells. AURKB, as a downstream key node of the FTO/SP1/AURKB pathway, could influence the tumor microenvironment of GC. The TCGA analysis showed that AURKB was negatively correlated with the Stromal score, which means that the AURKB high-expressing tumors had a relatively low degree of immune cell infiltration (Supplementary Fig. 6a). Immunocyte correlation analysis revealed that high expression of AURKB could cause decreased immune enrichment of B cells, T cells (CD4+ or CD8+), NK cells, DC, macrophages and CAFs (Supplementary Fig. 6b, c). More importantly, the analysis of tumor stemness indicated that tumor patients with higher expression of AURKB corresponded to stronger

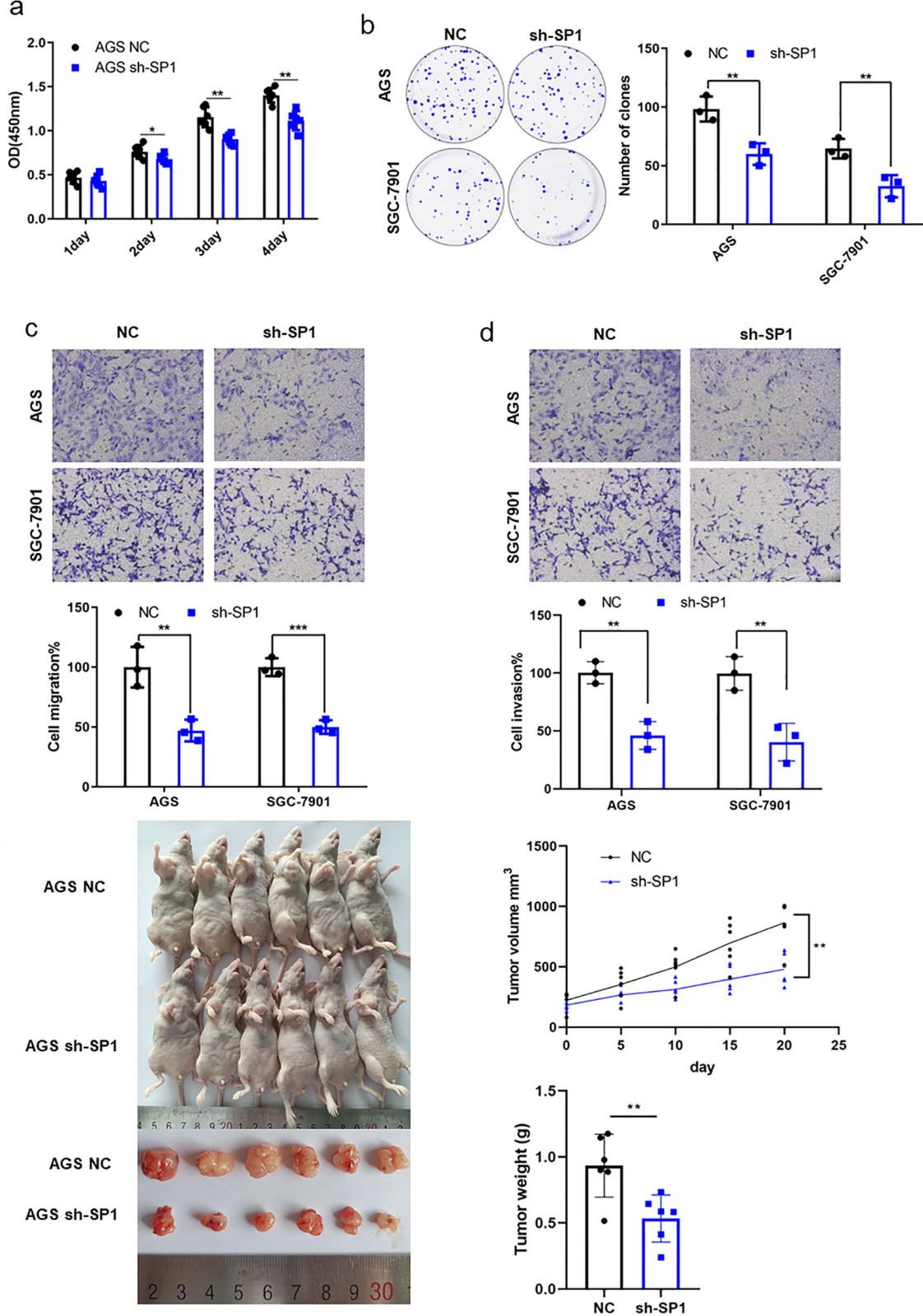

**Fig. 5 | SP1 knockdown inhibits the proliferation, migration, and invasion of GC cells in vitro as well as tumor growth in vivo. a** CCK-8 detected AGS cell proliferation. **b** Crystal violet staining marked the colonies of GC cell lines. **c, d** Transwell experiments tested cell migration (**c**) and invasion (**d**) abilities in SP1-knocked-down GC cells. **e** Tumor xenograft in nude mice revealed that SP1 knockdown suppressed tumor growth in vivo (up: tumor volume, down: tumor weight). *$p < 0.05$, **$p < 0.01$.

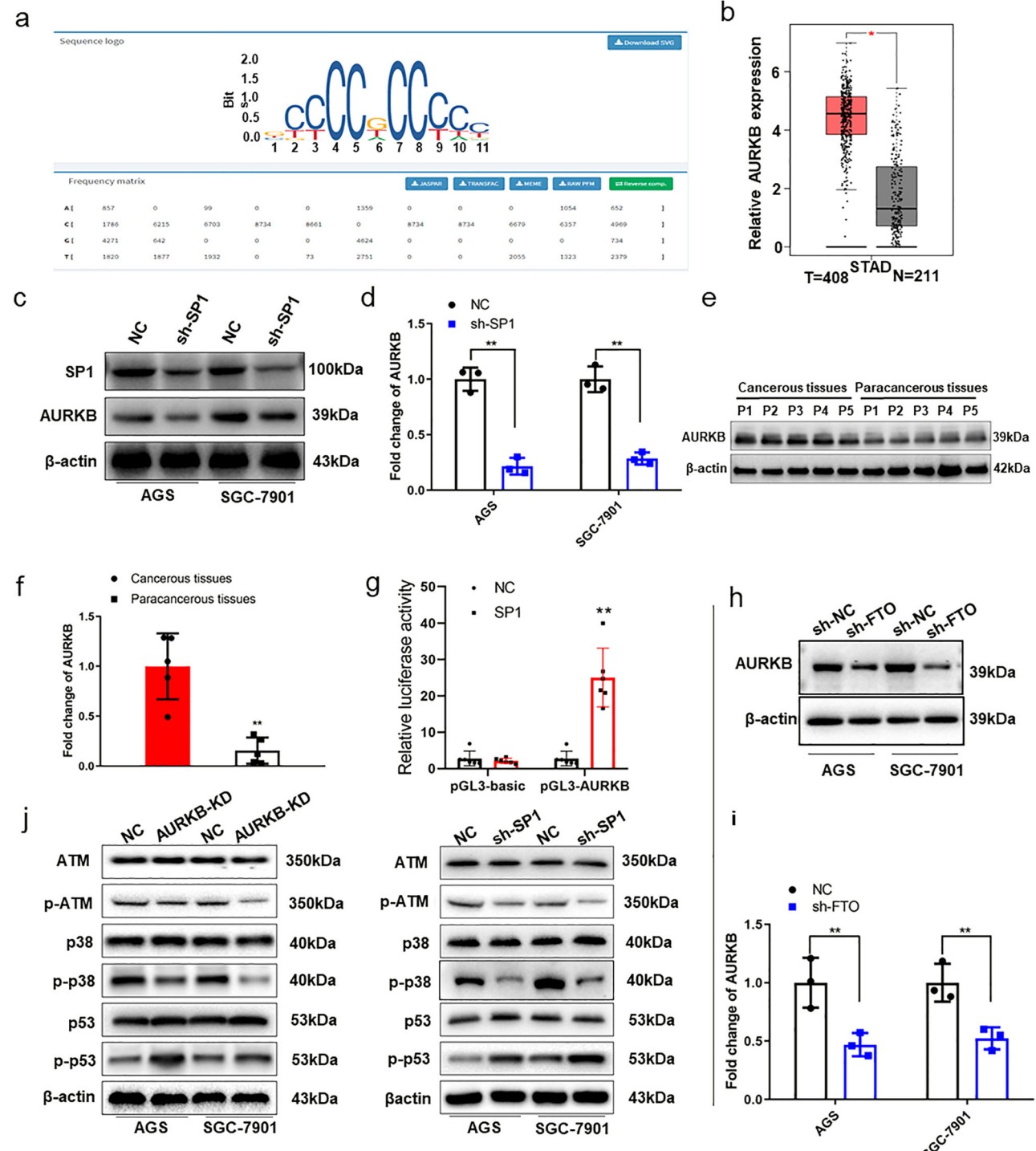

**Fig. 6 | *AURKB* is upregulated and serves as the target gene of SP1 in GC. a** The outcome of JASPAR website analysis shows the conserved sequences of SP1 binding site. **b** *AURKB* expression in STAD tissues from the TCGA database. **c, d** *AURKB* expression levels in SP1-knockdown GC cell lines in vitro. **e, f** AURKB expression levels in the clinical tissue samples of GC patients by Western blotting (**e**) and qPCR (**f**). **g** Dual-luciferase reporter gene activity of SP1 in *AURKB*-overexpressing AGS cells. **h, i** The effect of FTO-knockdown on AURKB expression in GC cell lines by Western blotting and qPCR. **j** The expression and activation of key downstream factors after *AURKB*- or SP1-knockdown. *$p < 0.05$, **$p < 0.01$.

stemness (Supplementary Fig. 6d). These findings suggested that enhanced expression of FTO/SP1/AURKB signaling in GC might also in-deeply control its malignant progression through regulation of tumor immune microenvironment, which could be very detrimental to the treatment and prognosis of patients. We also found that FTO enhanced the malignance of GC cells at least by upregulating SP1 expression and then AURKB, which in turn activated ATM and p38 by increasing their phosphorylation, resulting

in the deactivation of p53. P53 is a tumor suppressor that plays an important role in maintaining genomic stability and limiting abnormal cell proliferation. p38 phosphorylates and activates several transcription factors, including p53. Many studies have shown that p38 MAPK plays an important role in tumor cell invasion and metastasis through regulating EMT in cells[28,29]. McCarthy et al. reported that heparanase facilitates invasion and migration of GC cells probably through elevating phosphorylation

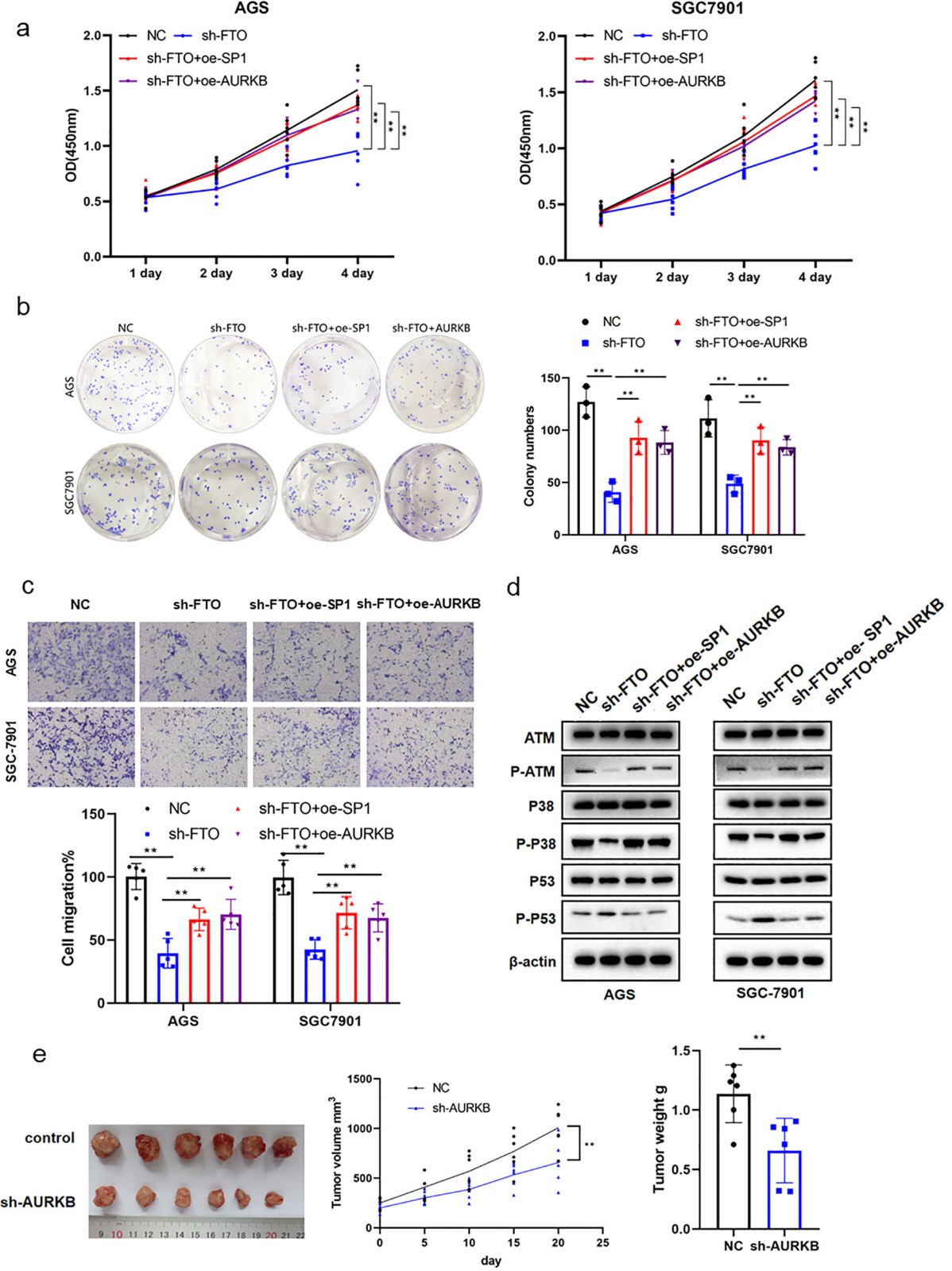

**Fig. 7 | FTO promotes GC cell progression via the *SP1-AURKB-ATM1* signaling axis. a** The viability of GC cells was detected by CCK-8. **b** Crystal violet staining marked the colons of GC cell lines. **c** The migration ability of GC was tested by Transwell without Matrigel. **d** The phosphorylated and total proteins of ATM, p38, p53 in GC cell lines after FTO knockdown with or without *SP1* and *AURKB*

overexpression (NC, sh-FTO, sh-FTO+oe SP1, sh-FTO+oe AURKB) were assessed by Western blot. **e** Tumor xenograft in nude mice revealed that *AURKB*-knockdown suppressed tumor growth (up) and tumor weight (down) in vivo. *$p < 0.05$, **$p < 0.01$.

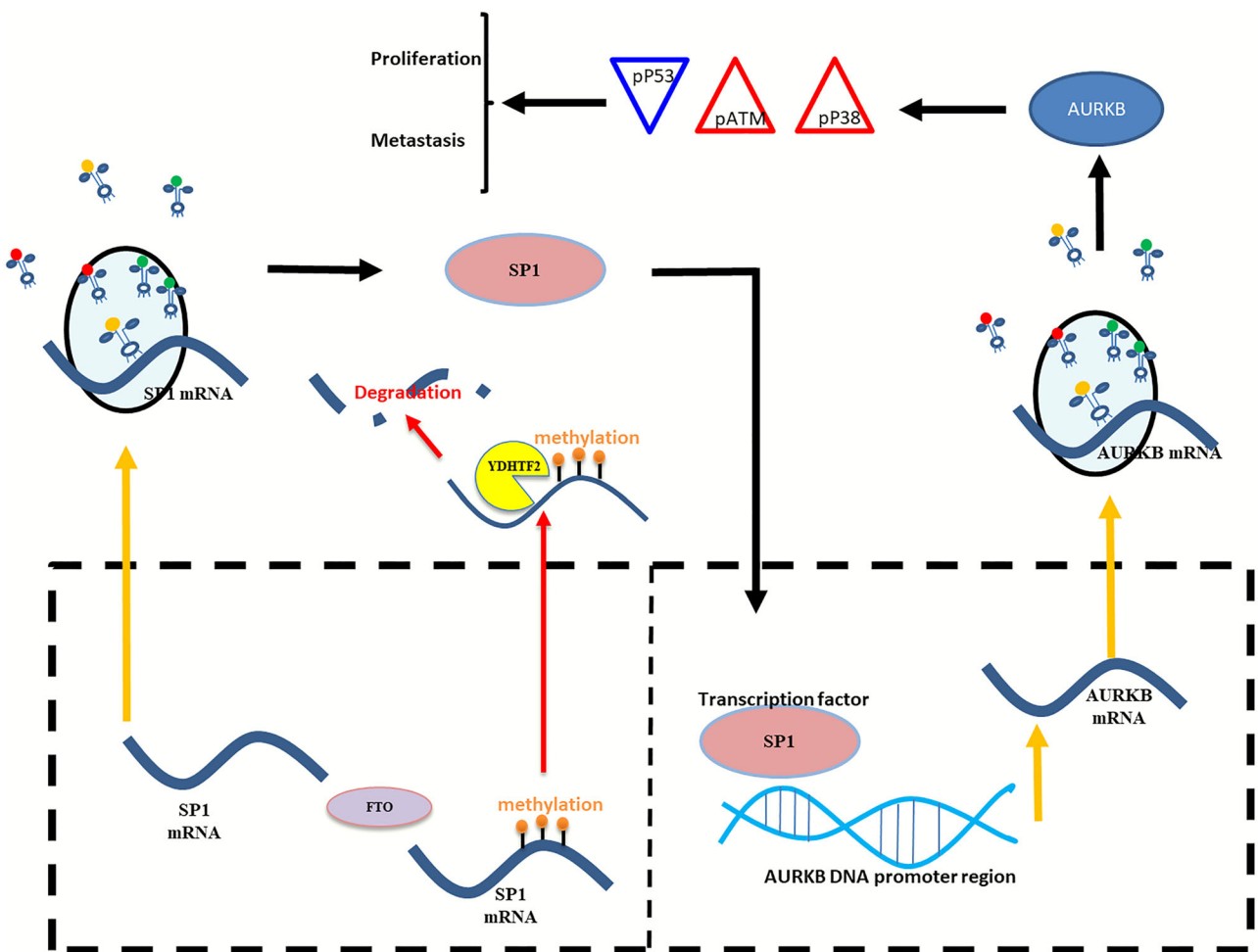

**Fig. 8 | Schematic of the function of the FTO-SP1-AURKB-ATM axis in GC progression.** The m6A demethylase FTO promotes GC tumorigenesis and progression by regulating the SP1-AURKB-ATM pathway.

of Src and p38[30]. Moreover, the study indicated that CLIC1 regulates gastric cancer cell migration and invasion via the ROS-mediated p38 MAPK signaling pathway[31]. The signaling pathways related to the two have extensive effects in the regulation of cell cycle and are a research hotspot for tumor therapy. This provides new insights into the underlying molecular mechanisms driving GC development and metastasis, suggesting that FTO may be a potential therapeutic target for GC treatment. Targeting FTO activity or expression could suppress tumor growth and metastasis by disrupting the oncogenic signaling axis. Our work also lays the foundation for developing FTO-specific inhibitors as novel therapeutics for GC treatment.

However, our studies primarily rely on in vitro experiments and xenograft mouse models to investigate the function of FTO in GC. While providing valuable cellular and molecular insights into GC progression, these approaches may not fully recapitulate the complex tumor microenvironment and interactions that occur in the human body. Therefore, it would be beneficial to validate the findings in more clinically relevant models, such as patient-derived xenografts or organoid cultures, which reflect the heterogeneity and complexity of human GC more faithfully.

In conclusion, our current work provides comprehensive evidence for the function of FTO in GC progression. We have elucidated that FTO promotes cell proliferation and metastasis by regulating the SP1-AURKB-ATM axis. The findings highlight the potential of FTO, SP1, AURKB, and ATM as prognostic biomarkers and therapeutic targets for GC treatment. This research broadens our understanding of the molecular processes underlying the development of GC and lays the foundation to develop innovative diagnostic tools and targeted therapies. While we identified the

SP1-AURKB-ATM axis as the major downstream pathway regulated by FTO, we cannot rule out the possibility that FTO also targets additional signaling pathways to facilitate GC progression. Further exploration of alternative downstream pathways and relevant cellular processes would enable us to properly comprehend the function of FTO in GC development.

## Methods
### Informed consent and patient specimen collection
The cancerous and paracancerous tissues were procured from patients diagnosed with GC at our institution during surgical treatments. All research procedures were approved by the Ethics Committee of the First Affiliated Hospital of Gannan Medical University according to the Helsinki Declaration. Written informed consent was obtained from each patient. The surgical tissue specimens were transiently frozen in liquid nitrogen and stored in an ultra-low temperature refrigerator at −80 °C[32].

### Data collection and analyzing
The RNA-Seq expression data and clinical pathological parameters of GC patients were downloaded from the Cancer Genome Atlas (TCGA) (http://cancergenome.nih.gov/) database. The gene expression and the survival rate were analyzed as described previously[33].

### Reagents
Fetal bovine serum (FBS) and RPMI-1640 medium were bought from Rockford (USA). TRIzol reagent (Invitrogen, USA). Magnesium RNA Fragmentation Module was obtained from NEB (USA). Secondary goat

anti-mouse-HRP and goat anti-rabbit-HRP antibodies were provided by Santa Cruz Biotechnology (USA). m6A-specific antibody (#202003) was purchased from Synaptic Systems (Germany). Antibodies anti- FTO (#14386),β-Actin(#4967), Sp1(#9389), ATM (#2873), p-ATM (#4526), p-p38 (#4511), p38(#9212), p-p53 (#), and p53(#2527) were provided by Cell Signaling Technology (USA). The primary antibody dilution concentration was 1:1000, and the secondary antibody dilution concentration was 1:5000. Matrigel was provided by Corning Life Science (USA). Lipofectamine 3000 transfection reagent and BCA Assay Kit were bought from Thermo Scientific (USA). PMSF and RIPA buffer were obtained from Beyotime (China). Anti-AURKB antibody (PA5-14075) and IGEPAL® CA-630 were obtained from Sigma-Aldrich (USA). Bestar qPCR MasterMix and qPCR RT kits were purchased from DBI Bioscience (China). PVDF membrane was provided by Sigma (USA). Gel Imaging System was bought from Bio-Rad (USA). Peroxide Block, ZytoChem Plus (HRP) Polymer Bulk Kit, DAB (diaminobenzidin) Substrate Kit, and EcoMount were bought from Zytomed Systems (Germany). A protease inhibitor cocktail was provided by Merck (Germany). Protein A Beads were obtained from GenScript (China).

### Cell lines and culture
Human gastric epithelial cells (GES-1) and human GC cell lines (HGC27, MKN-45, AGS, and SGC-7901) were bought from the Chinese Academy of Sciences (Shanghai, China). Human embryonic kidney 293 cells (HEK-293T) cells were obtained from Beina Chuanglian Biotechnology Institute (China). The cell lines, AGS, GSE-1, and HEK-293T, were maintained in the F-12K medium, while other cell lines were maintained in the RPMI-1640 medium supplemented with 10% FBS at 37 °C with 5% CO$_2$ and saturated humidity[34].

### RNA separation, library preparation, and sequencing
The separation and purification of total RNA were conducted using TRIzol reagent in accordance with the protocol provided by the vendor. Poly(A) RNA was cleaved by Magnesium RNA Fragmentation Module following isolation from 50 μg total RNA by Oligo-dT magnetic beads. Next, the RNA was cultured at 4 °C for 2 h with m$^6$A-specific antibody in IP buffer (0.5% Igepal CA-630, 750 mM NaCl, and 50 mM Tris–HCl) containing 0.5 mg/ml BSA, eluted and precipitated using 75% ethanol. Then the cDNA libraries were constructed with an average insert size of 100 ± 50 bp for the paired-end libraries. The paired-end sequencing (PE150) was performed on the Illumina Novaseq™ 6000 platform (LC-Bio, China) according to the manufacturer's protocol[35]. The primer and shRNA sequences are shown in Supplementary Table 2.

### Transfection
The Short hairpin RNAs (shRNAs) and NC shRNA were designed by GenePharma (China). The plasmid containing the target gene was designed and constructed by Obio Technology (China) and were transfected into GC cells by using Lipofectamine 3000 transfection reagent according to the manufacturer's instructions[36].

### RT-qPCR
The total RNAs were quantified using NanoDrop (Thermo Fisher Scientific, USA). A Bestar qPCR RT kit was used for cDNA synthesis. RT-qPCR was run with a 7500 Fast Real-Time PCR system (Applied Biosystems, USA). The $2^{-\Delta\Delta Ct}$ method was used to calculate the relative gene expression, as described previously[37].

### Western blotting
The total protein was extracted by RIPA buffer with PMSF (1 mM) separate and was quantified by BCA Assay Kit. Then, the lysates were subjected to 8–12% SDS-polyacrylamide gels for separation and transferred to PVDF membranes which were blocked with 5% skimmed milk for 1 h at RT and incubated with primary antibody overnight at 4 °C. After washing with TBST, the membranes were incubated with corresponding secondary antibody for 1 h at RT. Afterward, the blot was developed with ECL reagent and imaged by ChemiDoc Touch Imaging System[38]. All unprocessed scans of blots have been shown in Supplementary Fig. 7.

### Cell migration and invasion
Cell migration assay was performed using a 24-well Transwell (Corning Costar, Tewksbury, USA). Briefly, 500 μL of medium containing 10% FBS was added in the lower chamber and 300 μl of serum-free medium containing $2 \times 10^5$ cells was applied to the upper chamber with (for invasion) or without (for migration) Matrigel. After incubation for 24 h, cell migration or invasion was evaluated. The migrated or invasive cells were fixed with 4% paraformaldehyde, stained by crystal violet, and counted under a microscope (Leica DM4000, USA)[39].

### CCK-8 assay
The cells were incubated with CCK-8 reagent (10 μL/well) in a 96-well plate for 4 h at 37 °C, followed by measuring the absorbance at 450 nm by a microplate reader[40].

### Colony formation
GC cells were seeded into a 12-well plate at a density of 2000 cells per well and incubated for 10–14 days. The cells were fixed in 4% paraformaldehyde and stained by 0.5% crystal violet for 20 min. Finally, the numbers of colonies in each group were photographed and counted[41].

### Co-immunoprecipitation
The cells were homogenized in Co-IP lysis buffer and cell homogenate was centrifuged for 10 min at 12,000 × g. The supernatant was pre-treated with Protein A Beads (GenScript; L00273) for 20 min and then incubated with m$^6$A-specific antibody at 4 °C for 3 h. The interacting complex interacting with protein A beads was captured by agitation at 4 °C for 1 h. The immunoprecipitated proteins were collected by boiling the SDS-PAGE sample buffer after washing with PBS. Finally, the collected protein was applied for Western blotting assay[42].

### Methylated RNA immunoprecipitation assay
The GC cells were harvested and cross-linked followed by ultrasonic disruption in a lysis buffer at a low temperature. Proteinase K was used to remove proteins from the samples. Then, the RNA was extracted with phenol–chloroform and purified using a purification kit. Afterward, samples were reverse-transcribed with a reverse transcription kit[43]. The cDNA product was subjected to qPCR as described previously[37].

### RNA decay assay
RNA decay assay was performed by using actinomycin D (ActD) to verify RNA stability. The cells from 6-well plates were collected at 0, 3, and 6 h for qRT-PCR after treatment with 15 μg ActD. The relative gene expression levels at 0 h were normalized to 1 h[44].

### Dual Luciferase assay
The dual-luciferase reporter assay was performed to examine the interaction between the AURKB promoter and SP1. HEK-293T cells were co-transfected with pGL3-basic-AURKB-promotor and pLenti-CMV-SP1-3Flag-PGK-Puro as the subject, and the cells transfected with pGL3-basic-AURKB-promotor and pLenti-CMV-EGFP-3Flag-PGK-Puro vector were used as the control. The luciferase activity was assessed using the Luciferase Assay System according to the manufacturer's instructions (Promega, USA)[45].

### Xenograft model
The male BALB/c nude mice (6 weeks old) were bought from SLAC Laboratory (China). About $2 \times 10^6$ AGS cells in 100 μL DMEM/F-12 medium were injected subcutaneously into the right flank of each mouse for in vivo tumorigenicity assay. Tumor volumes were recorded every week. After 28 days, mice were sacrificed by euthanasia, and then the xenograft

tumors were stripped and weighed. The animal experimental protocol was approved by the Institutional Animal Care Utilization Committee of Gannan Medical University, and all animal experiments were performed according to the NIH Guide for the Care and Use of Laboratory Animals.

## Immunohistochemical (IHC) analysis and HE stain

Paraffin-embedded sections were routinely dewaxed, dehydrated in gradient and microwave antigen repair. Then, the sections were incubated with primary antibody for 12 h at 4 ℃ followed by staining with DAB and counterstaining with Gill's hematoxylin III. HE staining was performed using the HE staining kit (Solarbio, China). The preceding dewaxing and hydration processes were the same as the method of IHC[46].

## Statistics and reproducibility

The data were presented by mean ± standard deviation. Two-tailed Student's *t* test or one-way analysis of variance (ANOVA) test was used to assess the differences between groups. The Pearson correlation analysis was used for association evaluation. Kaplan–Meier method was used to analyze the association of prognosis in patients with gene expression levels. All statistical analyses were performed using by GraphPad Prism (La Jolla, USA) or R software (V 4.0.0). *p*-Value < 0.05 indicated a statistical significance[47].

## Data availability

The RNA sequencing results have been uploaded to the NCBI database (PRJNA1121210). Fig. 1a, Fig. 3d (Right), Fig. 4a, and Fig. 6b data were obtained from GEPIA. Fig. 1e and Fig. 4c data were obtained from Kaplan-Meier Plotter (https://kmplot.com/analysis/). Figure 3d (Left) data were obtained from TIMER. Figure 4b, Supplementary Fig. 1a, Supplementary Fig. 2a, and Supplementary Fig. 6 data were obtained from Sanger Box (http://www.sangerbox.com). Fig. 6a data were obtained from JASPAR. Figure 8 was created by the authors. Supplementary Fig. 2b data were obtained from STRING. Supplementary Fig. 2c data were obtained from SIGNOR. Supplementary Fig. 3b data were obtained from DAVID. Supplementary Fig. 3c data were obtained from KEGG Pathway (https://www.genome.jp/kegg/). Supplementary Fig. 4 and Supplementary Fig. 5 data were obtained from the TCGA database (https://www.cancer.gov/ccg/research/genome-sequencing/tcga). All other data are available from the corresponding author (or other sources, as applicable) on reasonable request

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

## Acknowledgements

This work was supported by *Project of Jiangxi Provincial Department of Health* (No. 202310837) and *The Innovation and Entrepreneurship Training Program for College Students in Jiangxi Province* (No. S202310413043).

## Author contributions

H.W. and B.L. designed the study. X.Z., Y.L., T.Z., W.L., W.H., X.T., and T.Y. performed the research. B.L. and Y.L. analyzed the data. X.Z. wrote the main paper text. All authors reviewed and approved the final paper. P.H., W.H., T.Y., and X.T. revised the paper.

## Competing interests

The authors declare no competing interests.
