## [Peer Review File · Communications Biology]

Reviewers' comments:

Reviewer #1 (Remarks to the Author):

Xueliang Zeng et al present a manuscript on the role of m6A modification in gastric cancer. They demonstrated the upregulation of FTO in gastric cancer and showed the prognostic relevance and the dependence of gastric cancer cell lines on FTO. Next, they identified SP1 as the gene most affected by FTO knockdown, and showed the importance of SP1 and its downstream, AURKB and ATM pathway. In general, the study is clearly presented, but I have some comments.

1. Regarding the figures using TCGA data, I'd like to know the relationship between the biomarkers identified (FTO/SP1) and the known gastric cancer phenotype. As histologic subtyping (diffuse type or intestinal type) or genomic subgroup (GS, CIN, EBV, MSI) already bear prognostic implication, the identification of the relationship can provide additional insight.
2. In line 298-300, the description is confusing. Did the authors suggest that FTO increased the methylation levels of SP1, and FTO knockdown decreased the expression levels of SP1?
3. As SP1 and AURKB are known to be related to the multiple function in cancer, it is difficult to directly attribute cancer phenotype or prognosis to the effect of m6A modification of a specific gene as a result of FTO activation. To comprehensively discuss the effect of FTO on cancer cells, the gene list with a variable m6A modification should be provided and the characteristic gene pathway should be investigated considering the uniqueness of the m6A modification data set the authors generated.
4. How do the authors mechanistically explain the augmentation of the migratory ability of cell lines by FTO/SP1/AURKB knockdown?

Reviewer #2 (Remarks to the Author):

In this paper, the gene expression and prognosis of patients were analyzed by bioinformatics, and then M6A sequencing, co-immunoprecipitation, Pearson correlation coefficient, luciferase activity and salvage experiments were used to study the role and mechanism of FTO in the development of GC in vivo and in vitro. finally, it was proved that M6A demethylase FTO promoted the occurrence and progression of gastric cancer by regulating SP1-AURKB-ATM pathway.

The following suggestions are available:

1. In the in vitro test to verify the relationship between FTO and GC cell proliferation, invasion and migration, the cell line used was relatively single.
2. In exploring the relationship between the expression of FTO and SP1 in cancer tissues and the prognosis of the patients, the tumor stage of the patients was not considered.
3. FTO gene knockout should increase the blank plasmid group in the verification of mRNA and protein level.
4. The culture environment of normal cells is different from that of tumor cells. Tumor cells have a unique tumor microenvironment, so we should try to simulate the tumor microenvironment in vitro.
5. Pay attention to the size and clarity of the text in the chart to improve typesetting.

Reviewer 1:

1, a statement describing the maximal tumour size/burden permitted by their ethics committee/IRB;

a statement confirming that the maximal tumour size/burden was not exceeded; in instances where it has been exceeded, justification should be provided.

Response: This is an important reminder! Animal ethics is a guideline that all researchers must follow. The AAALAC criteria are that the tumor burden of mice does not exceed 10% of body weight (Mice were aged 5 weeks and had a body weight of 20 ± 3 g). We double-checked the data and confirmed that the maximum mouse tumor weight was 1.176 g (Figure 5E) and 1.405 g (Figure 7E). All are within AAALAC limits. In addition, we include the ethics committee statement in the Supplementary Material.

2, Regarding the figures using TCGA data, I'd like to know the relationship between the biomarkers identified (FTO/SP1) and the known gastric cancer phenotype. As histologic subtyping (diffuse type or intestinal type) or genomic subgroup (GS, CIN, EBV, MSI) already bear prognostic implication, the identification of the relationship can provide additional insight.

Response: This is a very good advice! According to the reviewer's comment, we have analyzed the TCGA database and found that there was no difference in the expression of FTO/SP1 in different grades, stages and subtypes of gastric cancer. At the corresponding place in the revised manuscript, we added such descriptive sentence, "There was no significant relationship between expression of FTO (or SP1) and the grades, stages, or subtypes of gastric cancer". The additional data are presented in the Supplement (Fig S4 and S5).

Figure S4

Figure S5

3, In line 298-300, the description is confusing. Did the authors suggest that FTO increased the methylation levels of SP1, and FTO knockdown decreased the expression levels of SP1?

Response: Thank you for bringing this issue to our attention! We apologize for our

poor description which confuses you. We have corrected this description in the revised manuscript to make our findings clearer to the reader. “We found that RNA methylation level of SP1 was significantly increased in the FTO-knockdown group. Notably, transcriptome sequencing revealed that transcriptional level of SP1 was downregulated in the FTO-knockdown group. (Figure 3C). This result indicated that the m⁶A demethylases FTO significantly suppressed methylation of SP1 mRNA and enhanced transcription of SP1.”

4, As SP1 and AURKB are known to be related to the multiple function in cancer, it is difficult to directly attribute cancer phenotype or prognosis to the effect of m⁶A modification of a specific gene as a result of FTO activation. To comprehensively discuss the effect of FTO on cancer cells, the gene list with a variable m⁶A modification should be provided and the characteristic gene pathway should be investigated considering the uniqueness of the m⁶A modification data set the authors generated.

Response: This is a very constructive comment and enlightening to our further understanding of the effects of FTO activation and m⁶A modification on specific genes in cancer cells. In fact, a specific gene or protein is known to be related to the multiple intricate functions in cancers, it is difficult to directly and accurately attribute the special effect of a gene to the specific phenotypes of cancers. According to the reviewer’s direction, we have provided, in the Supplement section, the up-regulated and down-regulated genes with significant changes in RNA methylation sequencing and SP1 KEGG analysis. At the corresponding place in the revised manuscript, we added the description as below.

“The results of RNA methylation sequencing showed that, after FTO knockdown in GC cells, the methylation state of lots of genes especially the genes related to tumor metastasis and metabolism were significantly altered, indicating that FTO may play an obvious regulatory role in the occurrence and development of GC via regulating the homeostasis of RNA methylation and demethylation of malignant progression- or metabolism-related genes (Figure S3A-C). The included accurate genes or pathways in this process need further exploration following.”

5, How do the authors mechanistically explain the augmentation of the migratory ability of cell lines by FTO/SP1/AURKB knockdown?

Response: Thank you very much for your comment! Indeed, this is a comment well but difficult to respond. In fact, our data showed that, after FTO or SP1 or AURKB knockdown, the migratory or invading abilities of the GC cell lines were inhibited not augmented. Maybe our poor language or description made the poor readability, which confused the reviewer. In order to make our conclusion clearer to read and understand, in the revised manuscript, we have improved the corresponding mechanism exposition in Discussion sections. We really hope that the language level has been substantially improved in the revised manuscript.

“We also found that FTO enhanced the malignance of GC cells at least by upregulating SP1 expression and then AURKB, which in turn activated ATM and p38 by increasing their phosphorylation, resulting in the deactivation of p53. p53 is a tumor suppressor that plays an important role in maintaining genomic stability and limiting abnormal cell proliferation. p38 phosphorylates and activates several transcription factors, including p53. Many studies have shown that p38 MAPK plays an important role in tumor cell invasion and metastasis through regulating EMT in cells. McCarthy et al. reported that heparanase facilitates invasion and migration of GC cells probably through elevating phosphorylation of Src and p38. Moreover, the study indicated that CLIC1 regulates gastric cancer-cell migration and invasion via the ROS-mediated p38 MAPK signaling pathway. The signaling pathways related to the two have extensive effects in the regulation of cell cycle, and are a research hotspot for tumor therapy.”

Reviewer 2:

1, In the in vitro test to verify the relationship between FTO and GC cell proliferation, invasion and migration, the cell line used was relatively single.

Response: We agree! This is a very rigorous and scientific comment! It is known that a specific gene may also be differentially expressed in the same type of tumor cells. In

our study we found that, compared to normal cell line and tissue, FTO was differently high-expressed in GC cell lines and GC tissue. In order to verify the relationship between FTO and characteristic of GC cells, we currently selected AGS and SGC7901 cell lines with the highest expression of FTO (Figure 1G and 1H), and the functional experiments on proliferation, invasion and migration were performed after FTO knockdown (Figure 2C-F).

Thanks again! According to your suggestion, in our following studies we will select more FTO-high-expressing GC cell lines to extend the experiment, or employ GC cells with relatively FTO-low-expression to over-express FTO to reversely verify the relationship between FTO and of GC cells.

2, In exploring the relationship between the expression of FTO and SP1 in cancer tissues and the prognosis of the patients, the tumor stage of the patients was not considered.

Response: This is a very professional and important question. This is an important inspiration for our research! In response to this recommendation, we further analyzed the TCGA database, but unfortunately, there did not seem to be a significant difference between FTO and SP1 in patients with different stages of gastric cancer. At the corresponding place in the revised manuscript, we added such descriptive sentence, "There was no significant relationship between FTO expression and the grades of gastric cancer". The additional data are presented in the Supplement (Fig S4 and S5)

3, FTO gene knockout should increase the blank plasmid group in the verification of mRNA and protein level.

Response: We are very sorry for our carelessness caused this error, and thanks for your reminder. In the revised manuscript, we have re-worked the corresponding experiments, and supplemented the corresponding data (Figure 2B and 2C).

4, The culture environment of normal cells is different from that of tumor cells. Tumor cells have a unique microenvironment, so we should try to simulate the tumor microenvironment in vitro.

Response:

This is a professional, visionary and deep-in-soulful comment, we respectfully agree! Indeed, the tumors possess a unique microenvironment which control the fate of the tumor cells. Given the compositional and functional complexity of the tumor microenvironment, until now how to simulate the tumor microenvironment in vitro still a very difficult task. In recent years lots of scientists have tried to structure the tumor microenvironment in vitro to explore the real interaction of tumor cells with microenvironment components in vivo, such as, vascularization of tumor organoids(mini-tumor), co-culture of tumor cells with microenvironment cells/components, etc. However, these technologies are very complex, difficult, time-consuming, costly and half-baked. Given the current realities of our laboratory, we do not yet have the conditions, funding or capacity to accomplish these experiments. We will be happy to learn and try to construct the tumor microenvironment using tumor organoids and organoid microenvironmentalization techniques in vitro to expand our understanding of the issue in our following works, based on visionary and helpful comments from the reviewer.

In the revised manuscript, we additionally analyzed the relationship between tumor cells and the microenvironment using bioinformatics and pointed out this shortcoming of the in vitro experiments in the Discussion section. Described as follows.

“The tumors possess a unique microenvironment to control the fate of the tumor cells. AURKB, as a downstream key node of the FTO/SP1/AURKB pathway, could influence the tumor microenvironment of GC. The TCGA analysis showed that AURKB was negatively correlated with the Stromal score, which means that the AURKB high-expressing tumors had a relatively low degree of immune cell infiltration (Figure S6A). Immunocyte correlation analysis revealed that high expression of AURKB could cause decreased immune enrichment of B cells, T cells (CD4⁺ or CD8⁺), NK cells, DC, macrophages and CAFs (Figure S6B and S6C). More importantly, the analysis of tumor stemness indicated that tumor patients with higher expression of AURKB corresponded to stronger stemness (Figure S6D). These findings suggested that enhanced expression of FTO/SP1/AURKB signaling in

GC might also in-deeply control its malignant progression through regulation of tumor immune microenvironment, which could be very detrimental to the treatment and prognosis of patients.”

5, Pay attention to the size and clarity of the text in the chart to improve typesetting.

Response: Thank you very much for reminding us. We have modified the text in the figures and uploaded all the clearer figures.

In addition, we re-edited the text of our manuscript and corrected some spelling errors and inaccurate sentences, which are also highlighted in yellow in the revised manuscript.